# UNVEILING SIMPLICITIES OF ATTENTION: ADAPTIVE LONG-CONTEXT HEAD IDENTIFICATION

## ABSTRACT

The ability to process long contexts is crucial for many natural language processing tasks, yet it remains a significant challenge. While substantial progress has been made in enhancing the efficiency of attention mechanisms, there is still a gap in understanding how attention heads function in long-context settings. In this paper, we observe that while certain heads consistently attend to local information only, others swing between attending to local and long-context information depending on the query. This raises the question: can we identify which heads require long-context information to predict the next token accurately? We demonstrate that it's possible to predict which heads are crucial for long-context processing using only local keys: the core idea is to exploit a simple model for the long-context scores via second moment approximations. These findings contrast with earlier non-adaptive sparsifying schemes.

## 1 INTRODUCTION

The landscape of large language models (LLMs) is rapidly evolving, with modern transformer architectures capable of generating text from vast contexts. Recent advances have led to a significant increase in context window sizes, with Llama 3 (Dubey et al., 2024), DeepSeekv3 (Liu et al., 2024), and Gemini (Team et al., 2024a) supporting windows of at least 128k. When processing visual data, particularly videos, contexts tend to grow even longer, sometimes reaching millions of tokens (Liu et al., 2023a). However, long context modeling still poses significant challenges (Hsieh et al., 2024) with respect to both accuracy and the substantial cost of processing long contexts in terms of memory and run-time compute. In spite of their importance, our current comprehension of the attention mechanism in long-context tasks remains incomplete. This work aims to address some of these knowledge gaps.

Despite the overwhelming complexity of state-of-the-art models, certain simple behaviors in the attention mechanism are strikingly consistent. In particular, many forms of sparse behaviors have been observed, and exploited by numerous methods for efficient inference (see Section 6). Among them, Xiao et al. (2023) showed that even when computing the attention using only tokens close to the current token and some tokens close to the start of the context (which play the role of an attention "sink"), as illustrated on the left of Figure 1, the model is still capable of generating fluent text. We call such a collection of tokens (the last tokens and some of the first ones) a **local window** (see also Chen et al. (2024); Gu et al. (2024); Sun et al. (2024b)).

However, such a local window approximation, if applied to every attention head simultaneously, necessarily harms the capabilities of LLMs to retrieve and process long-context information (see e.g., Xiao et al. (2024)). Thus, we aim to identify the heads whose output can be well-approximated using a local window, and apply the approximation to those only. If a head can be approximated via a local approximation, we call it a **local head**, and otherwise it is a **long-context head**. In particular, we ask: Which heads can be approximated using a local window with minimal impact on downstream task performance?

Most existing approaches to this problem apply *static* criteria to label the heads – local vs long-context – once for all queries. Static criteria, as used by Xiao et al. (2024); Tang et al. (2024a), have the advantage that all key-value pairs (except for the few in the local window) of local heads can be discarded, thus saving memory. While recent works (Wu et al., 2024; Tang et al., 2024a; Hong

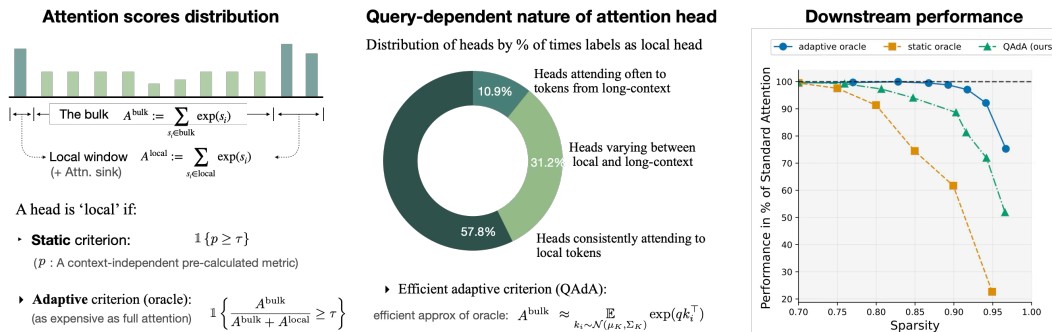

Figure 1: **Attention sparsity and its impact on efficiency.** *Left:* Attention scores are split into *bulk* ($A^{\text{bulk}}$) for distant tokens and *local window* ($A^{\text{local}}$) for nearby ones. A head is considered local if most of its attention mass falls within the local window. The static criterion pre-assigns local heads, while the adaptive oracle query-dependently compares bulk and local contributions but is computationally expensive. Our approximation models $A^{\text{bulk}}$ using a Gaussian distribution for efficiency. *Middle:* Oracle-based classification with $\tau = 0.6$ (see Figure 3 for the threshold) reveals three types of heads: consistently local (heads labeled more than $95\%$ of the times as local), often long-context (less than $50\%$), and varying, which switch behavior dynamically. *Right:* Comparison of three methods: Static (green) removes a fixed fraction of heads, the adaptive oracle (blue) dynamically selects heads but is costly, and the query-adaptive method (purple) achieves near-oracle performance with significantly lower cost. As sparsity increases, static pruning degrades performance, while the query-adaptive method remains robust. These results show that *most attention heads do not need to attend to the entire context*, enabling significant efficiency gains with *query-adaptive* head classification.

et al., 2024) provide some evidence that a *fixed* small subset of the heads are particularly relevant for processing long-context information, the following question remains unclear:

> *How much sparsity (measured as the average percentage of local heads) can we gain using query-adaptive criteria compared to static criteria?*

**Contribution 1.** We present an extensive analysis comparing a query-adaptive oracle criterion, which selects local heads independently for each token, with static criteria. We make two core observations, along with a series of minor findings regarding attention patterns: first, we find that static criteria can label up to 60% of the heads as local heads without impacting downstream task evaluations, which confirms the intuition from (Wu et al., 2024). Nevertheless, we find **that a query-adaptive oracle criterion can label a substantially higher percentage of heads as local heads (up to 90%) without sacrificing performance** (see Figure 1).

Unfortunately, the oracle requires the computation of the full attention scores. This leads to the following question:

> *For each query, can we determine which heads are long-context and which are local without computing the full attention scores?*

The relevance of this question is twofold: on one hand, answering it helps guide further research toward developing more compute-efficient attention mechanisms. On the other hand, it advances our understanding of the inner workings of attention mechanisms, which is central to mechanistic interpretability (see, e.g., Olsson et al. (2022); Cabannes et al. (2024)).

**Contribution 2.** We address this question by studying a novel query-adaptive attention criterion (QAdA) based on second-order statistics of the attention scores (briefly summarized in Figure 1). Our experiments on three families of LLMs, Llama (Dubey et al., 2024), Qwen (Bai et al., 2023) and Mistral (Jiang et al., 2023) applied to a variety of standard long-context benchmarks, as well as hard reasoning tasks embedded in long-context prompts, show that this relatively simple query-adaptive attention criterion can efficiently identify long-context heads: sparsity is increased at the cost of a smaller loss in downstream performance than for oracle static approaches. Hence, we demonstrate that **query-adaptive criteria can outperform static criteria in identifying long-context heads**. We also make several observations regarding attention patterns wrt. queries, layers and heads. These

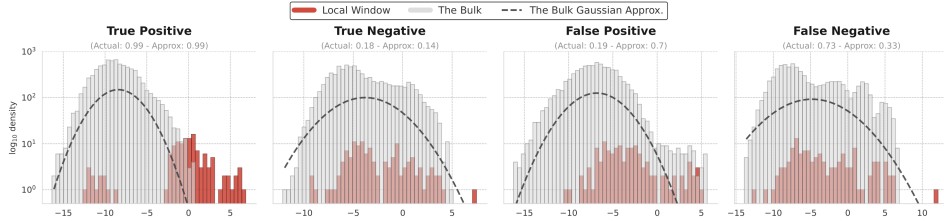

Figure 2: Examples of attention score distributions for each possible outcome with $\tau_{\text{approx}} = \tau_{\text{oracle}} = 0.6$ with the oracle criterion as ground truth. We show histograms of scores from the **local window** $\mathcal{I}$ (brown) and the **bulk complement** $[T] \setminus \mathcal{I}$ (gray), along with the bulk Gaussian approximation (black dashed line). The annotations above each plot indicate the values taken by the statistics used for the oracle criterion and the adaptive criterion.

findings pave the way for future research focused on developing the most efficient implementations of production-ready query-adaptive criteria.

## 2 PRELIMINARIES

We consider decoder-only transformer models (Vaswani, 2017), consisting of $L$-layers each containing one attention and one feed-forward block, using the rotary positional encoding (RoPE, Su et al. (2024)), which is commonly used in state-of-the-art open source LLMs, e.g., Llama3 (Dubey et al., 2024), Qwen (Bai et al., 2023) or Gemma (Team et al., 2024b). During inference, when predicting the next token, every single attention head takes as input a vector of (already rotated) queries $q \in \mathbb{R}^{1 \times d}$ and the (updated and rotated) cached key-value pairs $K, V \in \mathbb{R}^{T \times d}$, with sequence length $T$, and returns the weighted average of the values:

$$o = \text{softmax}(s)V \quad \text{with scores} \quad s = qK^\top / \sqrt{d} \tag{1}$$

**Local window approximation.** We are interested in long-context settings, where $T$ is large. For a given query and attention head, one can restrict the head's attention to a *local window*: instead of computing the head's attention scores with respect to each of the $T$ keys, only the attention scores corresponding to the first $T_{\text{sink}}$ input tokens (i.e. those closest to the start of the sequence) and the last $T_{\text{local}} - T_{\text{sink}}$ tokens are computed (as illustrated in Figure 1) and used to produce the output, where $T_{\text{local}}, T_{\text{sink}} \in \mathbb{N}$ are fixed parameters. Though they may not contain particularly relevant information, the first $T_{\text{sink}}$ tokens are included to serve as "attention sink" tokens, in line with the observations from Xiao et al. (2023). To summarize it more formally, we call $\mathcal{I} := \{1, \ldots, T_{\text{sink}}\} \cup \{T - T_{\text{local}} + T_{\text{sink}} + 1, \ldots, T\} \subset [T]$ the set of local indices, and the output of an attention head restricted to a local window is equal to $o_{\text{local}} = \text{softmax}(s_{\mathcal{I}})V_{\mathcal{I}}$, with $s_{\mathcal{I}} = qK_{\mathcal{I}}^\top / \sqrt{d}$.

**Query-adaptive oracle criterion** We want to define a criterion that adaptively labels, for each query, certain heads as **local heads** (for the given input token); we call the other heads **long-context heads**. Assuming that we have access to all scores, a natural way to define such a criterion is to compare the mass of attention scores from the local window $\mathcal{I}$ to some threshold. That is, given a threshold $\tau_{\text{oracle}}$, an attention head $h$, and its associated attention scores $s_i = qK_i^\top / \sqrt{d}, i \in [T]$, we define the **(query-adaptive) oracle criterion** $c_{\text{oracle}}^h$ which takes the head's scores $s$ as input:

$$c_{\text{oracle}}^h(s) = \mathbb{1}\left\{ \frac{\sum_{i \in \mathcal{I}} \exp(s_i)}{\sum_{i \in \mathcal{I}} \exp(s_i) + \sum_{i \notin \mathcal{I}} \exp(s_i)} \geq \tau_{\text{oracle}} \right\}. \tag{2}$$

If the criterion is satisfied for a given query, that is, if $c_{\text{oracle}}^h = 1$, the head *mostly attends* to tokens from the *local window*, and we call it a *local* head. On the other hand, if $c_{\text{oracle}}^h = 0$, the head assigns at least $1 - \tau_{\text{oracle}}$ attention mass to tokens from the global context, and we call it *long-context*. Note that our oracle criterion requires the computation of all the head's attention scores–as such, it is a tool of analysis, but it cannot be used as a practical way to increase compute efficiency.

## 3 IDENTIFYING LOCAL HEADS

Given that many attention heads swing between being local and being long-context depending on the input token (as illustrated in Figure 1 and further observed in Section 4.2), how can we identify local heads in a query-adaptive manner while only computing the attentions scores from the local window? Intuitively, such a criterion should distinguish between the two following cases:

- *Case 1 (long-context head):* The scores from the local window follow the same distribution as the remaining scores (second plot in Figure 2), and thus tokens from the local window cannot make up for most of the mass.

- *Case 2 (local head):* The scores from local tokens are significantly "out-of-distribution" on the right-sided tail (first plot in Figure 2). While this does not guarantee that the attention head assigns most of the mass to those tokens, as there might be outliers in the distribution of the non-local scores (third plot in Figure 2), this motivates us to label the head as a local head.

But how can we efficiently distinguish between the two cases? Our key insight is that a Gaussian approximation for the keys, which in turn yields a Gaussian approximation for the scores (black dashed line in Figure 2), provides a good approximation for deciding what is "in-distribution" (Case 1) and what is "out-of-distribution" (Case 2). Such an approximation in turn allows us to construct an efficient approximate version of the oracle criterion from Equation (2), that we call **query-adaptive attention (QAdA)**.

### 3.1 QUERY-ADAPTIVE CRITERION

The computational bottleneck in the oracle criterion from Equation (2) arises from the un-normalized mass $A^{\text{bulk}} := \sum_{i \notin \mathcal{I}} \exp(s_i)$ of the tokens from the bulk (see Figure 1). Let $\nu^{\text{bulk}}$ be the *empirical distribution* of the keys $k_i^\top$, $i \in [T] \setminus \mathcal{I}$ and let $T^{\text{bulk}} = T - T_{\text{local}}$. We can write the un-normalized mass as an expectation over $\nu^{\text{bulk}}$:

$$A^{\text{bulk}} = T^{\text{bulk}} \, \mathbb{E}_{k^\top \sim \nu^{\text{bulk}}} \exp\left(\frac{qk_i^\top}{\sqrt{d}}\right). \tag{3}$$

The main idea behind the proposed method is to now approximate $\nu^{\text{bulk}}$ by a product of Gaussians distributions with some mean $\mu_K$ and covariance $\Sigma_K$ (defined in Section 3.2):

$$\nu^{\text{bulk}} \approx \left(\mathcal{N}(\mu_K, \Sigma_K)\right)^{T^{\text{bulk}}}. \tag{4}$$

Such an approximation clearly does not apply at the level of individual keys. Indeed, according to the Gaussian approximation, all keys should be identically distributed, and this is definitely not the case as any two distinct keys store different positional information. Nevertheless, when averaged over the keys, we can hope that on a macro distribution level the approximation is accurate. More precisely, we propose to approximate:

$$\mathbb{E}_{k^\top \sim \nu^{\text{bulk}}} \exp\left(\frac{qk^\top}{\sqrt{d}}\right) \approx \mathbb{E}_{k^\top \sim \mathcal{N}(\mu_K, \Sigma_K)} \exp\left(\frac{qk^\top}{\sqrt{d}}\right). \tag{5}$$

In fact, the RHS can be computed in closed form. Indeed, we note that $\exp(qk^\top/\sqrt{d})$ follows a log-normal distribution:

$$\mathbb{E}_{k^\top \sim \mathcal{N}(\mu_K, \Sigma_K)} \exp\left(\frac{qk_i^\top}{\sqrt{d}}\right) = \mathbb{E}_{s \sim \mathcal{N}(\mu_s, \sigma_s^2)} \exp(s)$$
$$= \exp(\mu_s + \sigma_s^2/2) \tag{6}$$

with $\mu_s = q\mu_K^\top/\sqrt{d}$ and $\sigma_s^2 = q\Sigma_K q^\top/d$ the mean and variance of the scores. Assuming that we have access to the mean $\mu_K$ and covariance $\Sigma_K$ statistics (see further below), we can therefore compute an approximation of $A^{\text{bulk}}$ in constant run-time wrt. $T$!

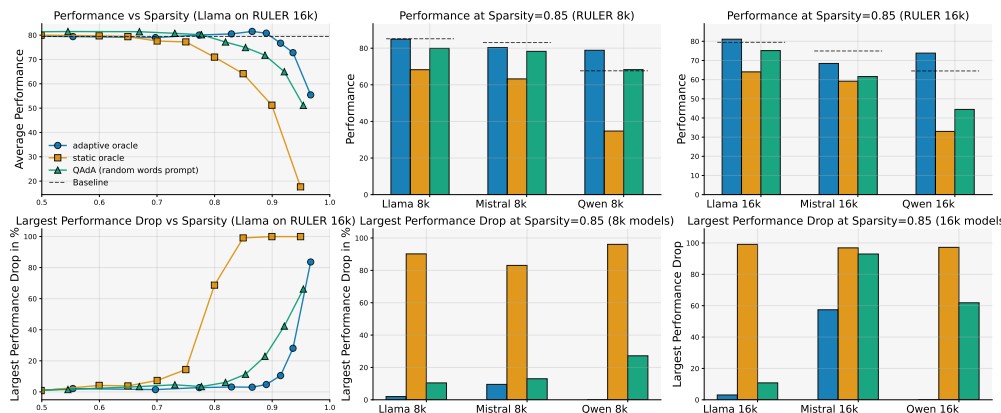

Figure 3: Comparison of QAdA against the adaptive and static oracles on the RULER benchmark. *Left:* For Llama 3-8B, we show the average performance (top) over the selected RULER 16k tasks as a function of the average sparsity for varying thresholds $\tau$, along with the worst-case performance drop (%) compared to the baseline performance among the selected tasks. *Middle and Right:* Average performance and worst-case drop for a fixed sparsity level of 0.85 across three model families—Llama, Mistral, and Qwen—on RULER 8k (center) and RULER 16k (right). The adaptive criterion consistently matches or outperforms the static oracle criterion, and in some cases (e.g., Mistral), even achieves performance comparable to the adaptive oracle. We provide more detailed results for each tasks and runs for a 32k context window in the Appendix.

In summary, given the moments $\mu_K$ and $\Sigma_K$, the query $q$ and the scores $s_i$ obtained from the local keys $k_i$, $i \in \mathcal{I}$, we propose to approximate the oracle criterion in Equation (2) via the following query-adaptive criterion (QAdA) with $A^{\text{local}} = \sum_{i \in \mathcal{I}} \exp(s_i)$:

$$c_{\text{approx}}^h(s) = \mathbb{1}\left\{ \frac{A^{\text{local}}}{A^{\text{local}} + T^{\text{bulk}} \exp\left(\mu_s + \sigma_s^2/2\right)} \geq \tau_{\text{approx}} \right\} \tag{7}$$

**Computing $\mu_K$ and $\Sigma_K$** To compute $\mu_K$ and $\Sigma_K$ and calibrate our criterion, we proceed as follows (see Appendix C for additional details): we let the model process a fixed sequence of *random words* of length similar to the contexts used in our evaluations. For a given layer and head, we let $\mu_K$ be the mean over the bulk tokens of the random sequence of the key vectors, i.e. $\mu_K = \frac{1}{T^{\text{bulk}}} \sum_{i \in [T] \setminus \mathcal{I}} K_i$. Similarly, we let $\Sigma_K = \frac{1}{T^{\text{bulk}}} \sum_{i \in [T] \setminus \mathcal{I}} K_i K_i^\top - \mu_K \mu_K^\top$. We have also considered other calibration methods, to which this one compares favorably - see our ablations in Appendix C.

**A remark on the (non-)optimality of the criterion** Our main objective is to empirically verify whether query-adaptive criteria can outperform static ones; as such, we do not pretend that the proposed criterion is optimally accurate or efficient. In particular, the chosen approximation of scores is only motivated by its computational convenience and by the fact that our experiments show that it leads to satisfactory performances; though approximating a distribution with a Gaussian distribution matching its first moments is a standard trick (as with PCA (Gewers et al., 2021)), we expect cleverer techniques to outperform it.

### 3.2 SUMMARY OF INFERENCE PIPELINE AND RUN-TIME COMPLEXITY

We describe how we apply the adaptive criterion studied in practice and explain how such a criterion could lead to decreased run-time complexity.

| criterion | criterion comp. | attention comp. |
|-----------|-----------------|-----------------|
| none | - | $O(Td)$ |
| oracle | $O(Td)$ | $O((1-\rho)Td + \rho T_{\text{local}}d)$ |
| QAdA | $O(T_{\text{local}}d + d^2)$ | $O((1-\rho)Td + \rho T_{\text{local}}d)$ |

Table 1: Run-time complexity of the oracle and adaptive criterion, as well as the cost of computing the resulting approximate attention. $\rho$ is the fraction of heads approximated by a local window of size $T_{\text{local}}$.

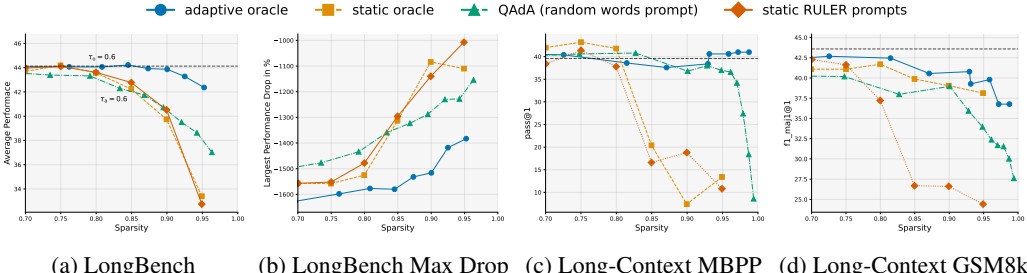

(a) LongBench     (b) LongBench Max Drop     (c) Long-Context MBPP     (d) Long-Context GSM8k

Figure 4: Similar to Figure 3, we show the average performance and largest performance drop for the LongBench benchmark, the pass@1 score for the MBPP task and the f1-score for the GSM8k task.

Before starting text generation, we calculate the moment statistics $\mu_K$ and $\Sigma_K$. Then, during decoding, before computing the attention output for a layer, we apply the query-adaptive criterion to every head in the layer, thus labeling a subset of them as local heads. We approximate the output of those using a local window, and compute the output of the others the usual way. We summarize the procedure in Listing 1 in the Appendix.

Unlike the oracle criterion from Equation (2), our query-adaptive criterion achieves a constant run-time complexity in $T$, assuming that $T_{\text{local}} \ll T$. Moreover, let $\rho$ be the fraction of times a head has been labeled as a local head and $d$ be the head dimension: then the average cost of computing the next token using the (approximated) attention mechanism is $O((1 - \rho)Td + \rho T_{\text{local}}d)$, as opposed to the $O(Td)$ operations required by the standard attention mechanism. These computations are summarized in Table 1.

## 4    EVALUATION ON DOWNSTREAM TASKS

### 4.1    EXPERIMENTAL SETTING

**Datasets.** We evaluate on the two standard long-context benchmarks, RULER (Hsieh et al., 2024) and LongBench (Bai et al., 2024). We also propose long-context variants of GSM8k (Cobbe et al., 2021) and MBPP (Austin et al., 2021), where we "hide" informative few-shot examples in a long-context prompt containing roughly $\approx 10k$ tokens. We refer the reader to Appendix B for further details.

**Models.** Our default model is the instruction fine-tuned *Llama 3-8B* model. We also use the two models *Mistral-7B-Instruct-v0.2* and *Qwen2-7B-Instruct* as provided by *HuggingFace*. To account for longer contexts, we set our models' RoPE parameter to $\theta = 2'000'000$, which is approximately the value from the NTK-aware interpolation (Peng & Quesnelle, 2023) for a context length of $32k$. For all evaluations, we choose a temperature of $0$, i.e. use the greedy decoding strategy. We always let $T_{\text{local}} = 128$ and use the first $T_{\text{init}} = 16$ tokens as "attention sink" tokens, leaving 112 tokens from the neighborhood closest to the current token (or sliding window).

**Methods** We implement the query-adaptive **oracle** criterion (Equation 2), alongside with two query-independent static criteria, **static oracle** and **static RULER**. The static RULER method, for a fixed sparsity threshold of $\alpha$ (we ablate over intervals of 5%), permanently labels as local the $\alpha$ percentage of heads that were most often labeled as local by the oracle criterion on prompts from the RULER tasks. The static oracle method, for a fixed sparsity threshold of $\alpha$, labels as local the $\alpha$ of heads that are most often labeled as local by the oracle criterion on the prompts of the processed task. See Appendix B for further details.

**Metrics** We use the standard metrics for evaluation provided by the corresponding benchmarks, which we refer to as the **performance**. For the LongBench benchmark, we compute the average normalized performance **(avg. norm. performance)**, which is obtained by dividing the performance by the performance of the standard full attention model. We always plot the performance as a function of the **sparsity**, that is the average percentage of heads labeled as local heads, and thus approximated

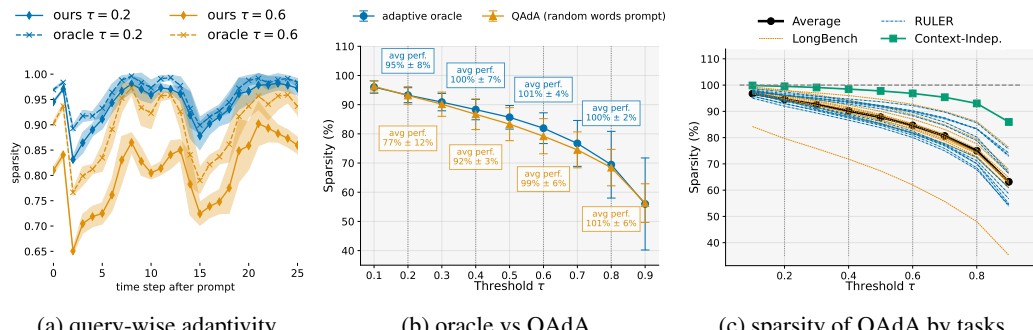

| (a) query-wise adaptivity | (b) oracle vs QAdA | (c) sparsity of QAdA by tasks |

Figure 5: **a)** The mean and standard deviation of the fraction of heads labeled as local heads as a function of time-steps for prompts from the "fwe" task. **b)** The average sparsity and standard deviation as a function of the threshold $\tau$ for Llama 3-8B over the RULER 8k and 16k, as well as the LongBench tasks. The annotations show the mean and standard deviation of the normalized performances (with $100\%$ being the performance of the standard dense attention). **c)** The average sparsities as a function of the threshold $\tau$, similar to those shown in b), are presented for each task, specifically for the QAdA criterion. Additionally, we present the average sparsity for a context-independent task. This task does not require context to be solved, and we observe that QAdA labels significantly more heads as local heads for the same threshold.

by a local window. For both adaptive and static criteria, the sparsity almost directly translates to a reduction of FLOPs used by the attention mechanism (minus a small constant overhead to compute the local scores).

## 4.2 PERFORMANCE ON RULER AND LONGBENCH

**Oracle gains over static.** We begin by comparing the adaptive oracle criterion against the static oracle criterion. We observe significant gains in performance across all models on the RULER benchmark in Figure 3, both in terms of the average performance, as well as the worst-case performance drop. The same observation also holds for the experiments on the LongBench benchmark in Figures 4a and 4b. For instance, for the Llama model we see a 20% increase in sparsity on the RULER tasks (from $\approx 70\%$ to $\approx 90\%$) and a $\approx 5 - 10\%$ increase on LongBench tasks at fixed performance level. These results underline the potential gains that are achievable by adaptive criteria for selecting attention heads over static ones. See also our additional experiments in Section F of the Appendix.

**QAdA outperforms static.** We observe that the adaptive criterion significantly outperforms the static criterion on the RULER task for sequence lengths of 8k in Figure 3, and also for lengths 16k for the Llama model. Moreover, the adaptive criterion matches the performance of the oracle static criterion and even slightly outperforms it on LongBench in Figure 4a and Mistral on RULER 16k. The only situation where we see performance drops compared to the static method is for Qwen on RULER 16k, where the score of the baseline model is itself very low. These results demonstrate that the criterion is capable of exploiting the query-adaptivity of attention heads.

**Outperforming the standard dense attention with Qwen** Finally, we observe in Figure 3 that both the oracle adaptive criterion and the adaptive criterion surpass the baseline performance of the standard full attention for Qwen on RULER $8k$ (see also Figure 15 in the Appendix). These gains are even more visible for the oracle criterion on RULER $16k$, where we find an average performance increase of more than 15 points for a sparsity of $0.85$. It is also worth noting that these gains are made possible by a query-adaptive approach and do not occur for static methods. These improvements highlight the fact that in long-context settings, models may attend to unnecessary parts of the context, which the query-adaptive criterion can effectively prune. Consequently, in such settings, query-adaptive criteria could provide benefits beyond computational efficiency, also leading to enhanced performance.

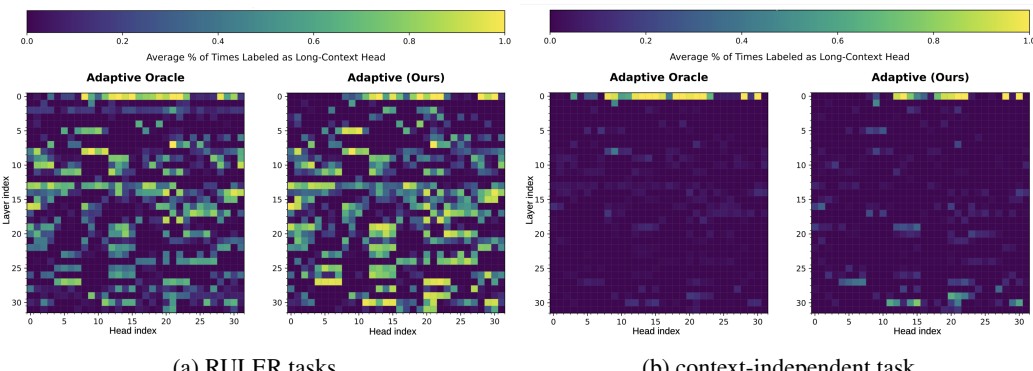

(a) RULER tasks          (b) context-independent task

Figure 6: We show for both the oracle adaptive and the adaptive criterion the % of times each head has been labeled as long-context head averaged over a) the six RULER 8k tasks with $\tau_{\text{oracle}} = \tau_{\text{approx}} = 0.6$ and b) the context-independent task based on the "qa-2" task from RULER.

## 4.3 PERFORMANCE ON REASONING AND CODE TASKS

While both the RULER and LongBench benchmarks require only short answers (sometimes less than 20 tokens), we also wonder how well the studied query-adaptive method is capable of selecting the "right" heads in challenging reasoning and code generation tasks, where the expected answers tend to be longer. We propose two long-context variants of the GSM8k and MBPP tasks (we provide examples in the Appendix) where we hide a few relevant few-shot examples in a mostly irrelevant long prompt. As instruction fine-tuned models do not require few-shot COT examples for solving the tasks, we instead use the pre-trained version of Llama 3-8B which heavily relies on these examples.

We show in Figure 4c and Figure 4d the performances on the long-context variants of the two tasks as a function of sparsity. We again observe that the adaptive criterion yields robust performance, outperforming the static criteria. Particularly striking are the gains for long-context MBPP, where both the oracle criterion and QAdA let us approximate almost all heads as local heads (more than 95%), while the performance of the static approaches significantly decreases beyond 80% sparsity.

## 5 DISCUSSION: ADAPTIVITY TO CONTEXTS

We saw in the previous section that QAdA is capable of selecting relevant heads for solving long-context tasks. In this section, we investigate which heads are selected by the model, and to what extent the model selects heads based on the context. Besides RULER and LongBench tasks, we also consider a *context-independent* task: more precisely, we take the context from the "qa-2" task from the RULER benchmark but replace the question with: *Can you describe LLMs in a few sentences?*. To solve this task, the model does not need to attend to the context. Overall, our experiments show that the query-adaptive criterion is also capable of *adapting to the context*.

**Query-wise sparsity.** First, we investigate whether QAdA is capable of changing the sparsity (average fraction of heads labeled as local heads) on a query-wise basis. We provide an illustrative example in Figure 5a, showing the average percentage of heads chosen by both the oracle and the adaptive criterion as a function of the time-step (query). We choose the "fwe" task, for which all responses to the prompts follow exactly the same pattern, and plot the mean and standard deviation as a function of the index of the generated token. We observe that the trend of the adaptive criterion aligns closely with the trend of the oracle criterion, and both vary strongly from token to token.

**Sparsity vs. Threshold.** We further plot in Figure 5b the average sparsity and the standard deviation of QAdA and the oracle criterion as a function of the threshold $\tau$. We make two findings: first, that QAdA closely follows the sparsity of the adaptive oracle criterion but tends to label slightly more heads as long-context. Second, that the standard deviation of the average sparsity (with respect to different tasks) is non-negligible, meaning that the sparsity can vary from task to task. This indicates that the adaptive criterion effectively adjusts the level of sparsity and is capable of adapting

to "difficult" tokens. Indeed, we further show in Figure 5c the average sparsities for each task for QAdA. We also plot in green the average sparsity when asking the model to generate a response for a task that does not require any knowledge from the context. As we can see, QAdA uses significantly fewer heads as long-context heads for this task than for the other tasks at the same threshold.

**Distribution of local heads across layers.** Finally, in Figure 6a and Figure 6b, we show the average percentage of times each head has been labeled as long-context for the RULER tasks and the context-independent tasks. For the RULER tasks, which require the model to look at the entire context, we see that both criteria show matching patterns and long-context heads occur across all layers. This demonstrates that our adaptive criterion successfully identifies long-context heads across all layers. Moreover, for the context-independent task, we see that while the first layer still attends to the full context, all other layers are essentially always approximated by the local windows.

## 6 RELATED WORKS

There is a large body of work studying and exploiting sparsity in attention heads. We refer the reader to (Wan et al., 2023; Zheng et al., 2024) for surveys and only discuss the most directly related works.

**Static classification of heads** Wu et al. (2024) showed that a few attention heads, called "retrieval heads," are particularly critical in retrieving long-context information, with multiple follow-up works (Tang et al., 2024a; Hong et al., 2024; Xiao et al., 2024; Cai et al., 2024; He et al., 2025). Most related to this paper is Xiao et al. (2024), which also proposed classifying the heads as long-context or local. All these methods statically assign labels to the heads before generation. They do so by analyzing attention patterns on selected tasks, or, as done in (Xiao et al., 2024), learning the assignment using gradient descent. Our paper crucially differs from these works as we explore the *query-adaptive* nature of attention heads to their contexts and do not require an additional dataset to label the heads.

**Query-adaptive sparsity.** Similar to this paper, there is an entire line of research that exploits query-dependent sparsity in some way. For instance, numerous works propose efficient approximations that retrieve per head the subset of tokens with the highest scores (Tang et al., 2024b; Ribar et al., 2023; Chen et al., 2021; Sun et al., 2024a). For context, multiple works also propose static variants that select the tokens for all queries (Zhang et al., 2023; Li et al., 2024; Oren et al., 2024). These works are **complementary** to this paper. More related to this paper is the approach taken by (Liu et al., 2023b; Akhauri et al., 2024), where a classifier is trained to dynamically predict which attention heads can be "dropped." The classifier takes as input the residual of an earlier layer and thus also adapts to the changing contexts. However, our paper crucially differs in two ways: first, we do not rely on any additional dataset for labeling the heads, nor do we require training an additional classifier. Second, we also distinguish between local and long-context heads, and do not simply drop heads.

## 7 LIMITATIONS

Though the studied adaptive criterion demonstrates that attention heads retrieve long-context information in a query-adaptive manner, which was one of our main goals, it does not constitute a production-ready tool. In particular, we have not studied the optimality of the Gaussian approximation used (in fact, we do not expect it to be optimal). We have not discussed efficient ways to turn query-adaptive head selection criteria into deployable attention mechanisms (e.g. FlashAttention-compatible implementations) either, as this topic warrants a separate article.

## 8 CONCLUSIONS

Our first key finding is that attention heads alternate between attending mostly to local tokens and being long-context on a token-wise basis. Our second key finding is that a simple test based on the second-order statistics of the keys and local scores, which we call QAdA (Query-Adaptive Attention), can efficiently predict this behavior: we proved it by testing QAdA on state-of-the-art models such as Llama, Qwen, and Mistral (7 to 8 billion parameters) and various important long-context benchmarks, including RULER and Longbench. These results further our understanding of attention mechanisms, and open the door to future research on more efficient sparsifying schemes.

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

```python
def adaptive_attention(q, k, v, mean_k, cov_k, Tl=128, log_thrs=0.6):
    mean_s = einsum('bhnd,hd->bhn', q, mean_k),
    var_s = einsum('bhnd,hde,bhne->bhn', q, cov_k, q)
    numerator = lse(q @ k[:,:, local_indices]/sqrt(d), dim=-1)
    log_bulk = log(seq_len - window_size) + var_s / 2 + mean_s
    denominator = lse(stack([numerator, log_bulk]),dim=0)
    mask = numerator - denominator > log(log_thrs)
    out[mask], out[!mask] = local_attn_(q, k, v,  mask), dense_attn_(q,
    k, v,  !mask)
    return out
```

Listing 1: Query-adaptive attention (QAdA) with local window approximation

Jiaming Tang, Yilong Zhao, Kan Zhu, Guangxuan Xiao, Baris Kasikci, and Song Han. Quest: Query-aware sparsity for efficient long-context llm inference. *arXiv preprint arXiv:2406.10774*, 2024b.

Gemini Team, Petko Georgiev, Ving Ian Lei, Ryan Burnell, Libin Bai, Anmol Gulati, Garrett Tanzer, Damien Vincent, Zhufeng Pan, Shibo Wang, et al. Gemini 1.5: Unlocking multimodal understanding across millions of tokens of context. *arXiv preprint arXiv:2403.05530*, 2024a.

Gemma Team, Thomas Mesnard, Cassidy Hardin, Robert Dadashi, Surya Bhupatiraju, Shreya Pathak, Laurent Sifre, Morgane Rivière, Mihir Sanjay Kale, Juliette Love, et al. Gemma: Open models based on gemini research and technology. *arXiv preprint arXiv:2403.08295*, 2024b.

A Vaswani. Attention is all you need. *Advances in Neural Information Processing Systems*, 2017.

Zhongwei Wan, Xin Wang, Che Liu, Samiul Alam, Yu Zheng, Jiachen Liu, Zhongnan Qu, Shen Yan, Yi Zhu, Quanlu Zhang, et al. Efficient large language models: A survey. *arXiv preprint arXiv:2312.03863*, 2023.

Wenhao Wu, Yizhong Wang, Guangxuan Xiao, Hao Peng, and Yao Fu. Retrieval head mechanistically explains long-context factuality. *arXiv preprint arXiv:2404.15574*, 2024.

Guangxuan Xiao, Yuandong Tian, Beidi Chen, Song Han, and Mike Lewis. Efficient streaming language models with attention sinks. *arXiv preprint arXiv:2309.17453*, 2023.

Guangxuan Xiao, Jiaming Tang, Jingwei Zuo, Junxian Guo, Shang Yang, Haotian Tang, Yao Fu, and Song Han. Duoattention: Efficient long-context llm inference with retrieval and streaming heads. *arXiv preprint arXiv:2410.10819*, 2024.

Zhenyu Zhang, Ying Sheng, Tianyi Zhou, Tianlong Chen, Lianmin Zheng, Ruisi Cai, Zhao Song, Yuandong Tian, Christopher Ré, Clark Barrett, et al. H2o: Heavy-hitter oracle for efficient generative inference of large language models. *Advances in Neural Information Processing Systems*, 36:34661–34710, 2023.

Zifan Zheng, Yezhaohui Wang, Yuxin Huang, Shichao Song, Mingchuan Yang, Bo Tang, Feiyu Xiong, and Zhiyu Li. Attention heads of large language models: A survey. *arXiv preprint arXiv:2409.03752*, 2024.

Yang Zhou, Hongyi Liu, Zhuoming Chen, Yuandong Tian, and Beidi Chen. Gsm-infinite: How do your llms behave over infinitely increasing context length and reasoning complexity? *arXiv preprint arXiv:2502.05252*, 2025.

## A  APPENDIX

## B  ADDITIONAL EXPERIMENTAL DETAILS

In this section we present additional details for the experiments.

**Additional details for the methods**    The best way to select the "right" subset of attention heads for the static criteria to which we compare is still widely understudied. In particular, it poses the fundamental challenge of which dataset should be chosen to select the heads in advance. Since we are primarily interested in how much query-adaptivity helps to improve, we compare against a **static oracle** criterion which uses the prompts on which the model is being evaluated to decide which heads to classify local heads. As this variant has "access to the test set", we expect it to be advantaged compared to other criteria. We also implement the **static RULER** method, which uses prompts from the RULER task for calibration. We present additional ablations for the choice of the static criterion in Figure 7. For both the static oracle and the static RULER method, and similarly to Wu et al. (2024); Tang et al. (2024a), we measure head patterns in a synthetic retrieval task, and select heads via the following simple **static criterion**:

- *Step 1*: Generate responses for selected prompts with dense attention. Compute the percentage of times each head would be labeled as local by the oracle criterion from Equation (2) with threshold $\tau_{\text{static}}$.

- *Step 2*: Calculate the $(1 - \alpha)$-quantile of these percentages across all heads $h$. Label heads below the threshold as *long-context* ($c_{\text{static}}^h = 0$) and those above as *local* ($c_{\text{static}}^h = 1$). These labels are now query-independent.

We refer the reader to Appendix C for further details regarding the computation of the moments used by **QAdA**.

**Choices for thresholds**    We ablate over the various thresholds $\tau_{\text{oracle}}, \tau_{\text{approx}} \in (0.1, 0.2, 0.3, 0.4, 0.5, 0.6, 0.7, 0.8, 0.9, 0.95, 0.99, 0.995)$, as well as $\alpha \in (0.05, 0.1, 0.15, 0.2, 0.25, 0.3, 0.35, 0.4, 0.45, 0.5, 0.55, 0.6)$ with $\tau_{\text{static}} = 0.6$. We ran additional ablations in Figure 7b for $\tau_{\text{static}}$ confirming that the choice $\tau_{\text{static}} = 0.6$ yields robust performance across all tasks.

**RULER tasks**    The RULER benchmark (Hsieh et al., 2024) consists of a collection of synthetic tasks with varying prompt sizes. These tasks are designed to challenge the model's capabilities in processing long-context information. We choose the two Q/A tasks, "qa-1" and "qa-2", the two aggregation tasks: common words extraction "cwe" and frequent words extraction "fwe", the variable tracing task "vt", and the multiquery needle-in-a-haystack task "niah". Especially, the two aggregation tasks "fwe" and "cwe" are known to be difficult baselines for achieving accuracy using efficient sparse attention mechanisms (see the discussion in Chen et al. (2024)).

**LongBench tasks**    The LongBench benchmark contains a selection of challenging real-world and synthetic tasks, including single-doc QA, multi-doc QA, summarization, and few-shot learning. We use a selection of tasks from the LongBench dataset for which the standard model achieves at least decent scores. We evaluate on the tasks: (Single-Document QA): "qasper", "multifieldqa-en", "multifieldqa-zh", "narrativeqa"; (Multi-Document QA): "2wikimqa", "musique", "hotpotqa"; (Summarization): "qmsum", "vcsum"; and (Few-shot Learning): "triviaqa".

**Long-context GSM8k and MBPP datasets**    In addition to the two standard benchmarks, RULER and LongBench, we also construct our own long-context tasks based on the reasoning task GSM8k (Cobbe et al., 2021) and the code-generation task MBPP (Austin et al., 2021). We use the standard evaluation protocol, but instead of using only the "correct" few-shot examples, we select 55 irrelevant few-shot examples in the same format generated from the SQUAD (Rajpurkar, 2016) dataset, as well as 5 relevant few-shot examples. We provide fragments of the example prompts below, with the relevant examples highlighted in green. The resulting context lengths are $\approx 10k$ for GSM8k and $\approx 11k$ for MBPP.

For these two tasks, we always use the pre-trained Llama3-8B parameter model (Dubey et al., 2024), instead of the instruction fine-tuned variant. The reason for choosing the pre-trained model is that the instruction fine-tuned model can solve these tasks without the need for few-shot examples, while the pre-trained model crucially depends on few-shot examples. Since these examples are hidden in a long context, the task becomes challenging, and the model needs to retrieve information from tokens far away in order to achieve high accuracy on the task.

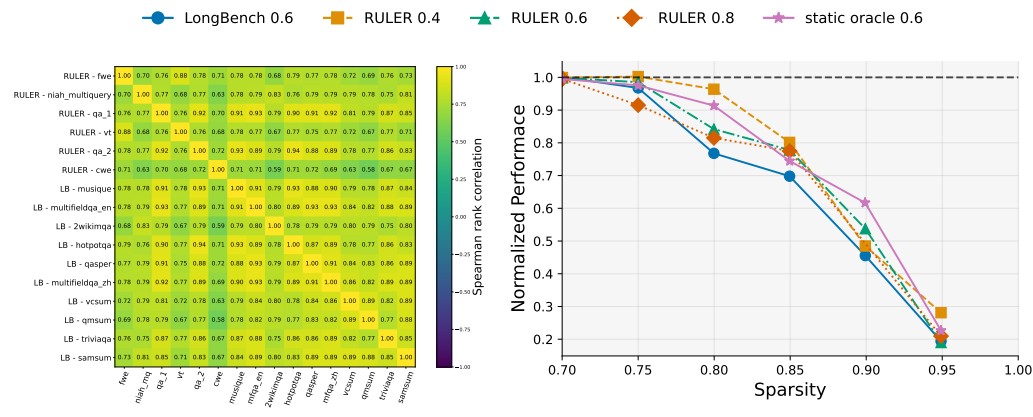

(a) Spearman rank correlation of heads

(b) Ablation over datasets for static criterion

Figure 7: **a)** The Spearman rank correlation of the attention heads ordered by the fraction of times they are labeled as Local Heads by the oracle criterion with $\tau = 0.6$. We see a high correlation among all tasks. b) Ablations for the static criterion using different datasets (LongBench, RULER and specific RULER task, called oracle) and threshold $\tau_{\text{static}}$ to label the heads. We use Llama3-8B on RULER 8k.

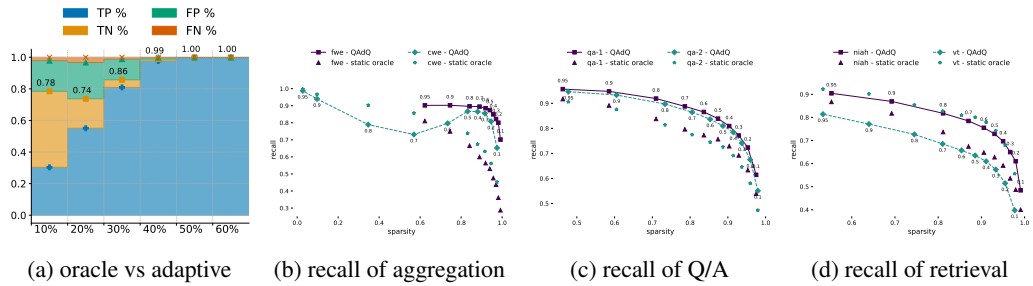

(a) oracle vs adaptive  (b) recall of aggregation  (c) recall of Q/A  (d) recall of retrieval

Figure 8: **a)** Accuracy and fraction of true/false positives/negatives for the 10% quantiles of the heads (labeled as local heads) for the adaptive criterion with $\tau_{\text{oracle}} = \tau_{\text{approx}} = 0.6$ on the RULER benchmark with sequence length 8k. **b,c,d)** The recall values of long-context heads selected by the oracle criterion for various thresholds $\tau_{\text{oracle}}$ when using the static and adaptive oracle criteria as a function of the average sparsity (percentage of local heads). We adjust the thresholds $\alpha$ (with $\tau_{\text{static}} = \tau_{\text{oracle}}$) and $\tau_{\text{approx}}$ to achieve matching sparsity levels. Annotations indicate the specific oracle thresholds $\tau_{\text{oracle}}$. We use Llama3-8B on RULER 8k.

## C  COMPUTING THE MOMENT STATISTICS

We describe in this section how to compute the moment statistics from Section 3 in greater details. We also consider several variants of our method, and conduct some ablations.

**Computing $\mu_K$ and $\Sigma_K$ from a sequence**  In all variants that we consider, the mean $\mu_K$ and covariance $\Sigma_K$ for a given head and layer are computed from the keys $(K_i)_{i \in [T]}$ (corresponding to the same head and layer) of some sequence $S$ of length $T$ as follows: we let as in the main text $[T] \setminus \mathcal{I}$ be the indices corresponding to the bulk of the sentence, and let

$$\mu_K = \frac{1}{T^{\text{bulk}}} \sum_{i \in [T] \setminus \mathcal{I}} K_i$$

and

$$\Sigma_K = \frac{1}{T^{\text{bulk}}} \sum_{i \in [T] \setminus \mathcal{I}} K_i K_i^\top - \mu_K \mu_K^\top.$$

We consider several different choices for $S$:

- A *random word prompt* created by permuting words from a Wikipedia article (including HTML syntax). This is the method used in the main text.

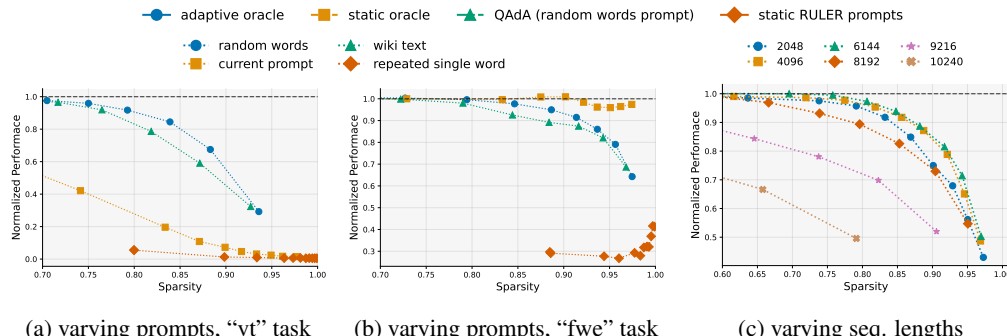

(a) varying prompts, "vt" task   (b) varying prompts, "fwe" task   (c) varying seq. lengths

Figure 9: Ablations for the content of the prompt (e-f) and the length of the prompt (g) used to generate the mean $\mu_K$ and covariance $\Sigma_K$ for the adaptive criterion from Section 3. We show the normalized performance as a function of sparsity (e) for the "vt" task and (f) for the "fwe" task and (g) averaged over the RULER 8k tasks, respectively.

- A *wiki prompt* by concatenating Wikipedia articles.

- A *single words prompt* repeating the word "observation."

- The *current prompt*, i.e. the prompt on which the method is currently being evaluated.

For each of these and unless otherwise specified, we let the sequence be of roughly the same length *minus 1024* as the prompt on which the method is evaluated, as suggested by the results of our ablations below. For grouped-query attention (Ainslie et al., 2023), as used by Llama, we utilize the same moments for each query within the group, as these heads share the same keys.

### C.1 ABLATIONS OVER THE CHOICE OF PROMPT

We present ablations for the choice of the prompt used to generate the mean $\mu_K$ and variance $\Sigma_K$ statistics.

**Prompt.** We ablate in Figures 9a-9b and 10 over the content of the prompts used to generate the moments statistics. We show the curves only for the two illustrative RULER tasks "variable tracing" ("vt"), that has a highly repetitive structure, and "frequent word extraction" ("fwe"). Maybe surprisingly, we find for the "vt" task that the best performance is attained when using randomly sampled words, while repetitively using the same words results in the worst performance. Moreover, using the exact moments (i.e., *current prompt*) also results in very poor performance. This is not the case for the "fwe" task, where using the current prompt achieves the best performance. We believe that the failure on the "vt" task is explained by the repetitive structure of the prompt, which resembles the structure of the repeated single word prompt that also yields very poor performance. In summary, we find that although using "current prompt" can sometimes yield strong performance ("fwe") task, it is not robust to the choice of task. In contrast, "random words prompt" using a distinct dataset yields more robust performance, which is why we focus on this method in the main text.

**Sequence Length.** We compare in Figure 9c the performances of our query-adaptive method using the *random words prompt* for different lengths of the prompt used to generate the mean $\mu_K$ and covariance $\Sigma_K$. We show the average normalized performance across all RULER 8k tasks. We see drastic drops in performance when the prompt used to compute the statistics gets longer than the length of the prompt on which the method is evaluated (that is $\approx 8100$ tokens long), whereas performance is surprisingly robust to variations for shorter sequence lengths. This dependence to the length of the random words prompt suggests that while the statistics $\mu_K$ and $\Sigma_K$ do not contain any information about the task (as we use random words), they nevertheless contain positional information critical for the criterion to identify the right set of local heads.

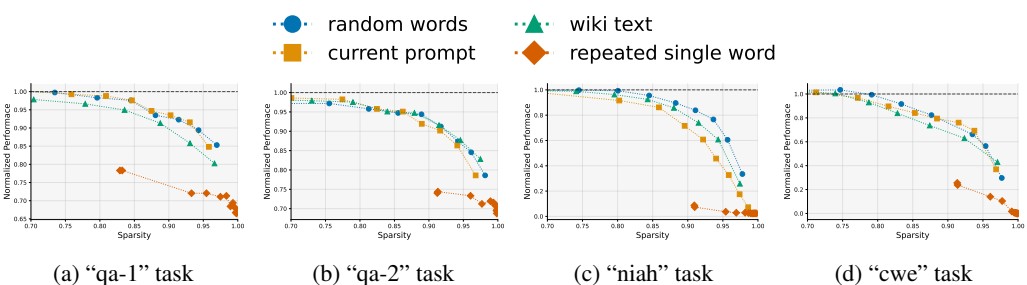

| (a) "qa-1" task | (b) "qa-2" task | (c) "niah" task | (d) "cwe" task |

Figure 10: Ablations for varying prompts. Same as Figure 9a and 9b for the additional RULER 8k tasks using Llama 3-8B.

## D  RECALL OF ATTENTION HEADS

In this section, we analyze how well our adaptive criterion from Section 3 can recall the heads selected by the oracle criterion; in other words, how effectively it serves as a proxy for the oracle. Here we use the current prompt to generate the moment statistics.

**Accuracy**  We generate responses using standard dense attention and store the scores used to compare the two criteria using the current prompt to generate the moments. For each task, we group the heads into $10\%$ quantiles based on the percentage of times the oracle criterion has been satisfied. For each quantile (averaged over the six selected RULER tasks), we show the fraction of true positives, true negatives, false positives, and false negatives, where a true positive means that both the oracle and adaptive criteria labeled a head as a local head.

We find that the adaptive criterion always correctly identifies the top $50\%$ of the heads that are consistently local heads. Moreover, we find even higher accuracies for the lower quantiles where heads vary between local and long-context. Interestingly, we see that the false negative rate is much lower than the false positive rate for these heads. As a result, the adaptive criterion selects fewer heads than the oracle criterion. This observation is counter-intuitive to the observations made in Section 5, where we observed that our adaptive criterion tends to select more heads than the oracle criterion for the same threshold. The explanation here is that in this section we compare the criterion on scores obtained when using standard full attention. This is necessary to allow a direct comparison between the two criteria. In contrast, in Section 5 we compare the average sparsity when using the approximate attention that approximates all labeled heads by a local window.

**Recall of long-context heads.**  We further compare our adaptive criterion with the oracle static criterion in their ability to identify long-context heads selected by the oracle criterion. We show in Figure 8b-8d the recall value of long-context heads selected by the oracle criterion for different oracle thresholds $\tau_{\text{oracle}}$ as a function of the sparsity (fraction of heads labeled as local heads by the oracle criterion). To allow for a direct comparison between static and adaptive, we choose $\tau_{\text{approx}}$, resp. quantile $\alpha$ (with $\tau_{\text{static}} = \tau_{\text{oracle}}$), such that the average sparsity is the same as the one of the oracle criterion. We plot the curves for all (selected) RULER tasks, and find that our test achieves consistently a higher recall value than the oracle static assignment (except for the "vt" task, for which the *current prompt* choice for the moments breaks down, as discussed in Section C.1).

## E  DISCUSSION: GAUSSIAN APPROXIMATION

In this section, we further discuss the Gaussian approximation exploited by our criterion in Section 3.

**Approximation error**  We measure the approximation error arising from Equation (5). We show in Table 2 the average log difference $|\log A^{\text{bulk}} - (\log(T^{\text{bulk}}) + \mu_s + \sigma_s^2/2)|$ (first row) between the un-normalized mass of the bulk and our Gaussian approximation from Equation (5). Taking the exponent, we find that the Gaussian approximation is typically off by a factor of $\approx 2 - 5$, and thus clearly imprecise. In comparison, in the third row, we show the same statistics, when replacing the scores by i.i.d samples from a Gaussian distribution with matching mean and variance. This error

| Method | all $\mu \pm \sigma$ | top 20% $\mu \pm \sigma$ | top 10% $\mu \pm \sigma$ |
|---|---|---|---|
| | RULER 8k task "fwe" | | |
| Log error | $0.41 \pm 0.58$ | $0.50 \pm 0.98$ | $0.57 \pm 1.27$ |
| Dist. local | $3.44 \pm 1.73$ | $1.78 \pm 1.38$ | $1.54 \pm 1.23$ |
| Gaussian opt. | $0.15 \pm 0.18$ | $0.14 \pm 0.21$ | $0.15 \pm 0.25$ |
| | RULER 8k task "Q/A-2" | | |
| Log error | $0.37 \pm 0.52$ | $0.63 \pm 0.75$ | $0.74 \pm 0.83$ |
| Dist. local | $2.80 \pm 1.55$ | $1.17 \pm 0.98$ | $1.29 \pm 1.08$ |
| Gaussian opt. | $0.18 \pm 0.22$ | $0.25 \pm 0.34$ | $0.29 \pm 0.40$ |

Table 2: The mean and standard deviation for the terms log difference $|\log A^{\text{bulk}} - (\log(T^{\text{bulk}}) + \mu_s + \sigma_s^2/2)|$ (Log error) and $|\log A^{\text{bulk}} - \log A^{\text{local}}|$ (Dist. local) for all heads (first column) and the 20% and 10% percentiles of heads most often labeled as local heads by the oracle criterion with $\tau_{\text{oracle}} = 0.6$. We further show the "Log error" when replacing the scores by i.i.d. Gaussian samples instead with matching mean and variance. This indicates the achievable error assuming that the Gaussian approximation holds true. We use Llama3-8B on RULER 8k.

captures the "optimal" error under a Gaussianity assumption. As we can see, this error is significantly smaller.

Nevertheless, we are practically interested in whether the Gaussian assumption suffices to make an accurate prediction on whether the head is a local or long-context head. To that end, we also compare in the second row the average log difference $|\log A^{\text{bulk}} - \log A^{\text{local}}|$. Indeed, if this distance is much larger than the average log error arising form the Gaussian approximation, we expect our criterion to nevertheless be accurate. As we observe, this is the case. Taking again the exponent, we find that the $A^{\text{bulk}}$ and $A^{\text{local}}$ typically differ by factors around $\approx 15 - 50$. Interestingly, however, we see that the gap becomes more narrow when only considering the top 20% (resp. 10%) of heads most frequently selected by the oracle criterion as long-context heads. Finally, we also show the average standard deviation.

# F  ADDITIONAL EXPERIMENTS

**Performances for individual tasks**  We showed in Figures 3 and 4a the aggregated performances over the tasks. For completeness, we further show in Figures 11-18 the performances for the individual tasks. We further also show the performance of QAdA applied using the *current prompt* (see Section C).

**Longer context-windows**  The reason for selecting context window sizes of length below 20k is that, although current models are often still capable of retrieving needles, their performance on more challenging tasks such as common word extraction massively drops when moving beyond 16k context windows, even before the application of any sparsifying method such as ours (see appendix of Hsieh et al. (2024)). Recent benchmarks Modarressi et al. (2025); Zhou et al. (2025) further underline this observation, demonstrating that most LLMs fail to solve simple tasks well before reaching a 32k context length. Such difficult tasks are the focus of this paper, and precisely the tasks where existing approaches for sparse attention mechanisms fail, which have been discussed more in depth e.g. in Chen et al. (2024). As such, we find experiments on context lengths above 32k rather uninformative. Nevertheless, we include in Figure 17 results for a context window of size 32k.

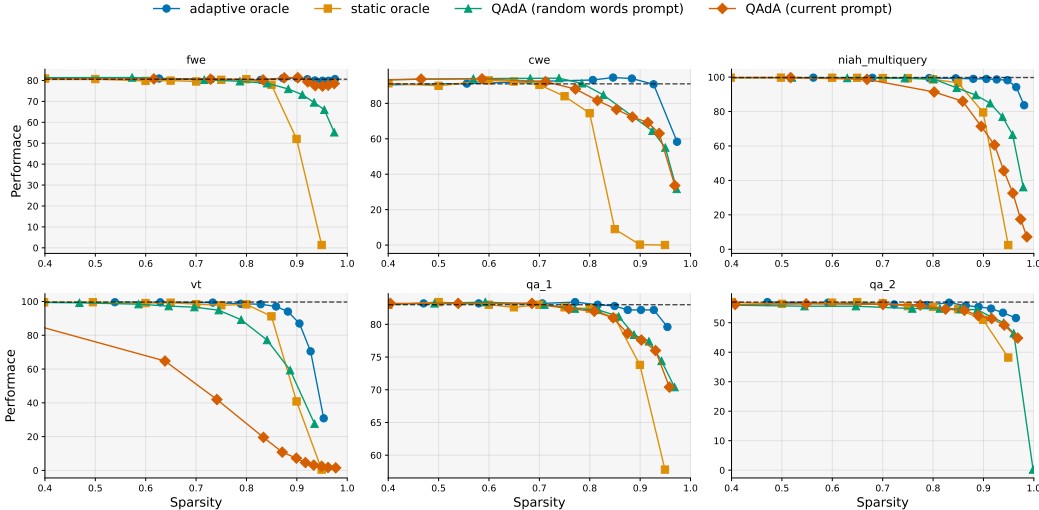

Figure 11: Performances for individual tasks for RULER 8k using Llama-3 8B as in Figure 3
.

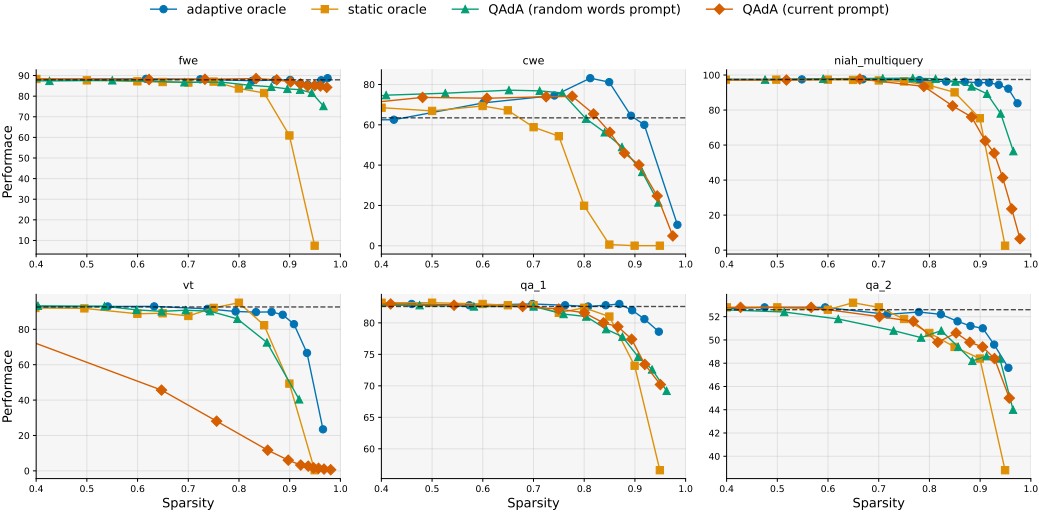

Figure 12: Performances for individual tasks for RULER 16k using Llama-3 8B as in Figure 3
.

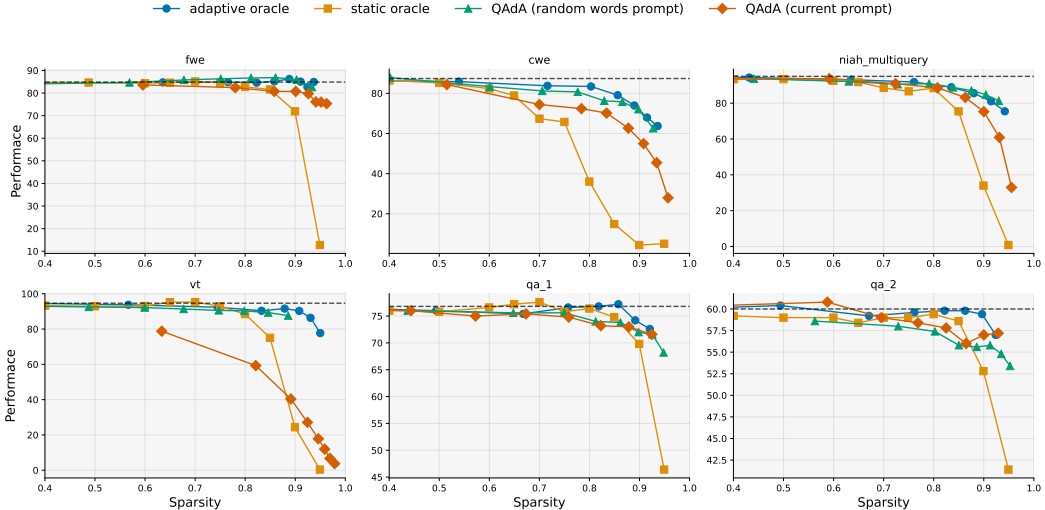

Figure 13: Performances for individual tasks for RULER 8k using Mistral-7B as in Figure 3
.

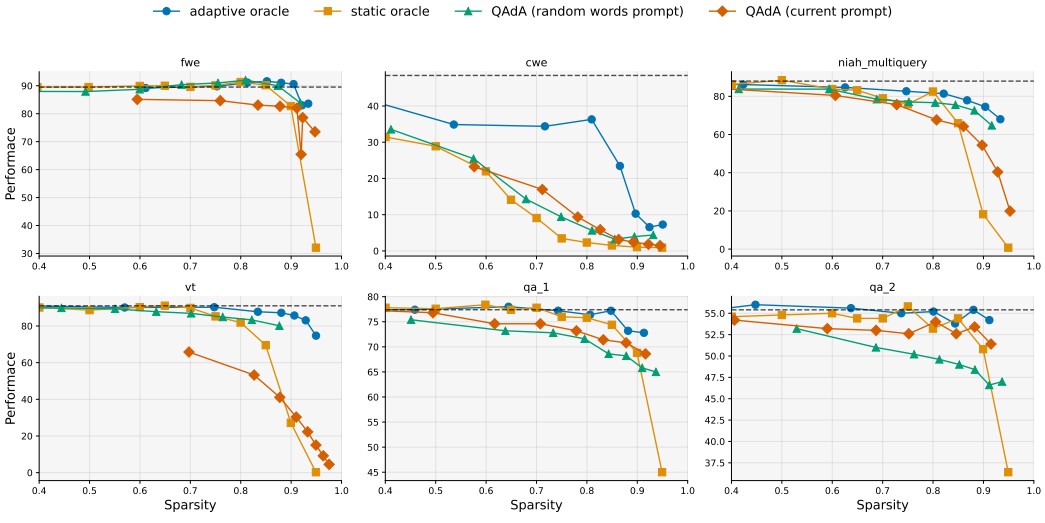

Figure 14: Performances for individual tasks for RULER 16k using Mistral-7B as in Figure 3
.

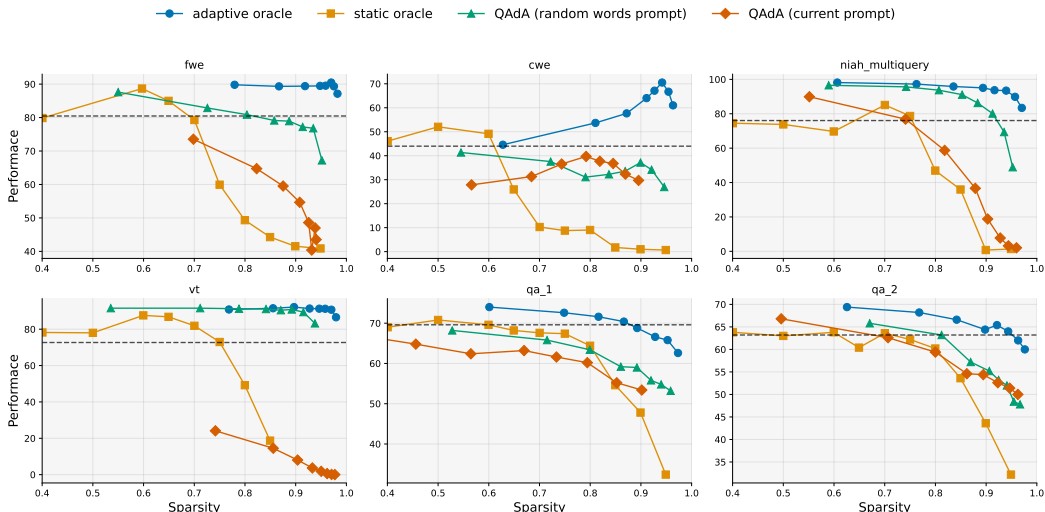

Figure 15: Performances for individual tasks for RULER 8k using Qwen-7B as in Figure 3
.

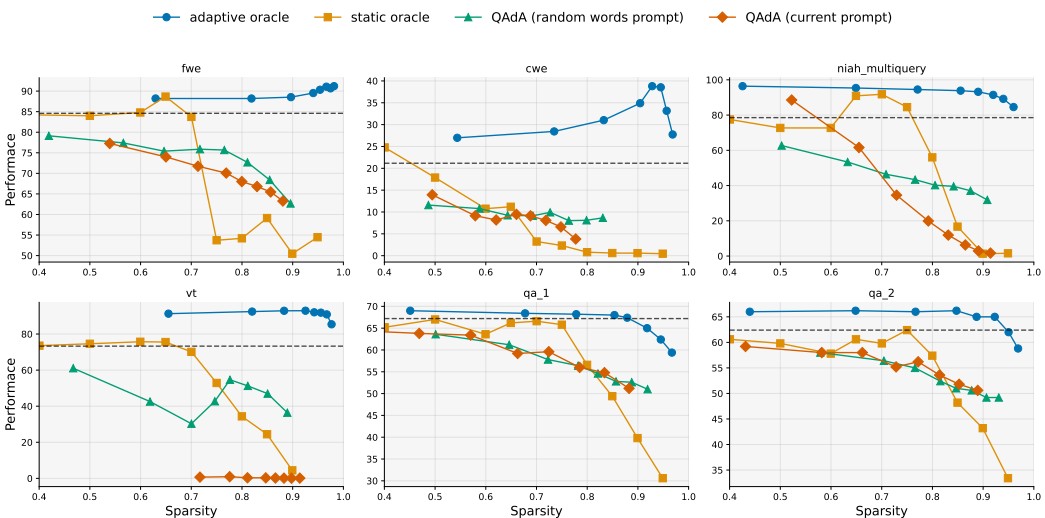

Figure 16: Performances for individual tasks for RULER 16k using Qwen-7B as in Figure 3
.

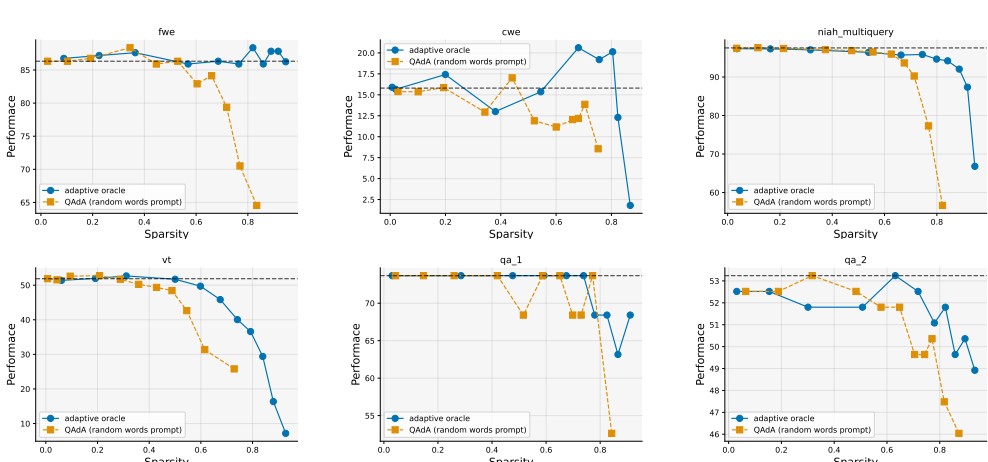

Figure 17: Performances for individual tasks for RULER 32k using Llama-3 8B as in Figure 3
.

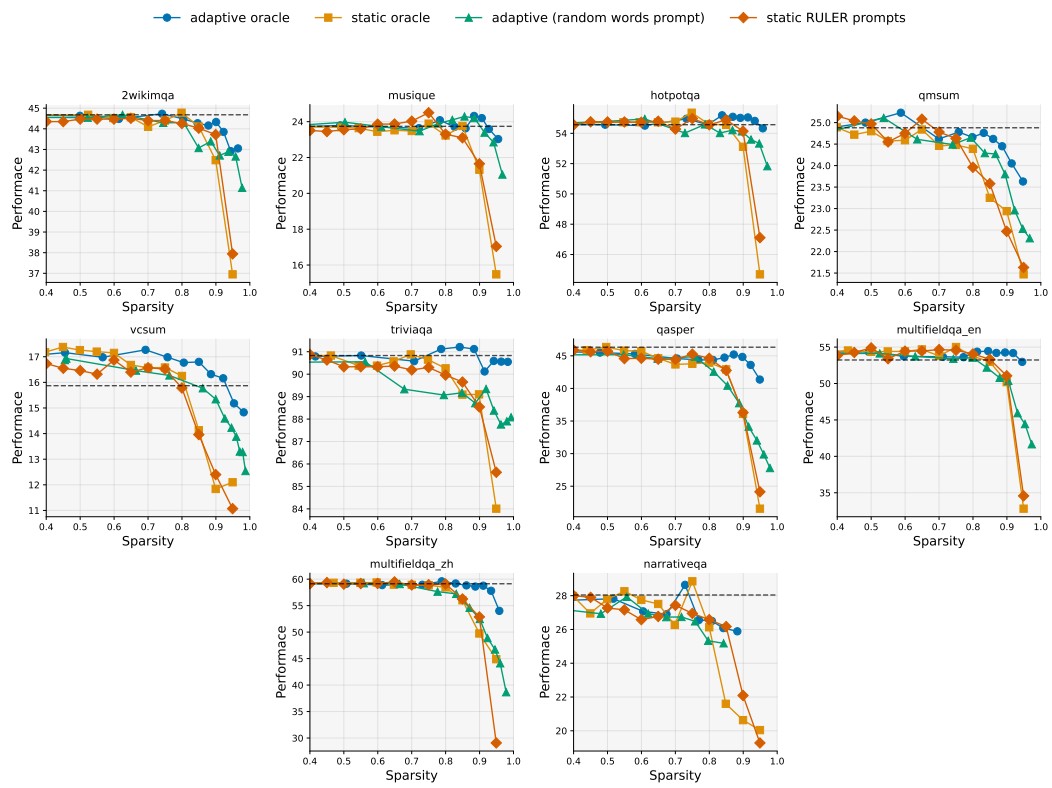

Figure 18: Performances for individual tasks for LongBench as in Figure 4a

## Example Prompt for long-context MBPP

[...]

Q: Due to extreme variation in elevation, great variation occurs in the climatic conditions of Himachal . The climate varies from hot and subhumid tropical in the southern tracts to, with more elevation, cold, alpine, and glacial in the northern and eastern mountain ranges. The state has areas like Dharamsala that receive very heavy rainfall, as well as those like Lahaul and Spiti that are cold and almost rainless. Broadly, Himachal experiences three seasons: summer, winter, and rainy season. Summer lasts from mid-April till the end of June and most parts become very hot (except in the alpine zone which experiences a mild summer) with the average temperature ranging from 28 to 32 °C (82 to 90 °F). Winter lasts from late November till mid March. Snowfall is common in alpine tracts (generally above 2,200 metres (7,218 ft) i.e. in the higher and trans-Himalayan region).
What is the climate like?
A: varies from hot and subhumid tropical The answer is varies from hot and subhumid tropical.

Q: James decides to buy a new bed and bed frame. The bed frame is $75 and the bed is 10 times that price. He gets a deal for 20% off. How much does he pay for everything?
A: The bed cost 75*10=$750
So everything cost 750+75=$825
He gets 825*.2=$165 off
So that means he pays 825-165=$660 The answer is 660.

Q: Liz sold her car at 80% of what she originally paid. She uses the proceeds of that sale and needs only $4,000 to buy herself a new $30,000 car. How much cheaper is her new car versus what she originally paid for her old one?
A: If Liz needs only $4,000 to buy a new $30,000 car, that means she has $30,000-$4,000=$26,000 from the proceeds of selling her old car
If she sold her car at 80% of what she originally paid for and sold it for $26,000 then she originally paid $26,000/80% = $32,500 for her old car
If she paid $32,500 for her old car and the new one is $30,000 then, the new one is $32,500-$30,000 = $2,500 cheaper The answer is 2500.

Q: Unlike in multicellular organisms, increases in cell size (cell growth) and reproduction by cell division are tightly linked in unicellular organisms. Bacteria grow to a fixed size and then reproduce through binary fission, a form of asexual reproduction. Under optimal conditions, bacteria can grow and divide extremely rapidly, and bacterial populations can double as quickly as every 9.8 minutes. In cell division, two identical clone daughter cells are produced. Some bacteria, while still reproducing asexually, form more complex reproductive structures that help disperse the newly formed daughter cells. Examples include fruiting body formation by Myxobacteria and aerial hyphae formation by Streptomyces, or budding. Budding involves a cell forming a protrusion that breaks away and produces a daughter cell.
What are produced in cell division?
A: two identical clone daughter cells The answer is two identical clone daughter cells.

Q: Janet's ducks lay 16 eggs per day. She eats three for breakfast every morning and bakes muffins for her friends every day with four. She sells the remainder at the farmers' market daily for $2 per fresh duck egg. How much in dollars does she make every day at the farmers' market?
A:

## Example Prompt for long-context GSM8k

[...] You are an expert Python programmer, and here is your task: Due to extreme variation in elevation, great variation occurs in the climatic conditions of Himachal . The climate varies from hot and subhumid tropical in the southern tracts to, with more elevation, cold, alpine, and glacial in the northern and eastern mountain ranges. The state has areas like Dharamsala that receive very heavy rainfall, as well as those like Lahaul and Spiti that are cold and almost rainless. Broadly, Himachal experiences three seasons: summer, winter, and rainy season. Summer lasts from mid-April till the end of June and most parts become very hot (except in the alpine zone which experiences a mild summer) with the average temperature ranging from 28 to 32 °C (82 to 90 °F). Winter lasts from late November till mid March. Snowfall is common in alpine tracts (generally above 2,200 metres (7,218 ft) i.e. in the higher and trans-Himalayan region).
What is the climate like? Your code should pass these tests:
empty
[BEGIN]
varies from hot and subhumid tropical
[DONE]

You are an expert Python programmer, and here is your task: Write a function to find the similar elements from the given two tuple lists. Your code should pass these tests:
assert similar_elements((3, 4, 5, 6),(5, 7, 4, 10)) == (4, 5)
assert similar_elements((1, 2, 3, 4),(5, 4, 3, 7)) == (3, 4)
assert similar_elements((11, 12, 14, 13),(17, 15, 14, 13)) == (13, 14)

[BEGIN]
def similar_elements(test_tup1, test_tup2):
res = tuple(set(test_tup1) & set(test_tup2))
return (res)
[DONE]

You are an expert Python programmer, and here is your task: Unlike in multicellular organisms, increases in cell size (cell growth) and reproduction by cell division are tightly linked in unicellular organisms. Bacteria grow to a fixed size and then reproduce through binary fission, a form of asexual reproduction. Under optimal conditions, bacteria can grow and divide extremely rapidly, and bacterial populations can double as quickly as every 9.8 minutes. In cell division, two identical clone daughter cells are produced. Some bacteria, while still reproducing asexually, form more complex reproductive structures that help disperse the newly formed daughter cells. Examples include fruiting body formation by Myxobacteria and aerial hyphae formation by Streptomyces, or budding. Budding involves a cell forming a protrusion that breaks away and produces a daughter cell.
What are produced in cell division? Your code should pass these tests:
empty
[BEGIN]
two identical clone daughter cells
[DONE]

You are an expert Python programmer, and here is your task: Write a python function to remove first and last occurrence of a given character from the string. Your code should pass these tests:
assert remove_Occ("hello","l") == "heo"
assert remove_Occ("abcda","a") == "bcd"
assert remove_Occ("PHP","P") == "H"

[BEGIN]

