# OpenReview forum: "Unveiling Simplicities of Attention: Adaptive Long-Context Head Identification"
_ICLR.cc/2026/Conference — Submitted to ICLR 2026_

### Official Review · Reviewer_JQs7 · 2025-10-31

**Soundness:** 2
**Presentation:** 2
**Contribution:** 2
**Rating:** 4
**Confidence:** 3

**Summary:**

The paper proposes Query-Adaptive Attention (QAdA), motivated by the finding that applying local window approximations to all heads degrades long-context reasoning and that head behavior varies across queries. QAdA identifies, per query, which heads require long-range attention by comparing local and bulk contributions through a Gaussian approximation of key statistics, without any additional training. Experiments on RULER and LongBench with Llama-3, Mistral, and Qwen show that QAdA achieves near-oracle performance while maintaining high sparsity and offering insight into query-dependent head behavior.

**Strengths:**

**1. Simple yet effective method without additional training**

The proposed Gaussian-based criterion operates without retraining or auxiliary data and provides a lightweight yet robust mechanism to identify query-specific long-context heads.

**2. Intersection between head selection and query-adaptive sparsity**

The paper connects two previously separate directions of research, static head classification and token-level adaptive sparsity, and formulates head-level query adaptivity as a unified framework.

**3. Meets oracle performance in empirical evaluation**

Experiments on RULER and LongBench with Llama-3, Mistral, and Qwen models show that QAdA achieves near-oracle accuracy while maintaining high sparsity and interpretability.

**Weaknesses:**

**1. Comparison limited to oracle and static baselines**

The paper evaluates QAdA only against a static criterion and an oracle reference. It does not include existing adaptive sparsity or head classification methods as baselines. Because there is no direct comparison with prior work, it is unclear whether QAdA actually improves upon existing techniques or simply presents an alternative formulation of similar ideas.

**2. Complementarity with query-adaptive sparsity not empirically validated**

The paper claims that QAdA is complementary to query-adaptive sparsity, but this claim is not supported by experiments. The paper does not test a combined setting to show whether the two methods work better together. Without such evidence, the claim remains speculative, and the approaches may overlap rather than reinforce each other.

**3. Conceptual contribution is modest and primarily integrative**

The work unifies two familiar directions, head selection and query-adaptive sparsity, within a single framework. While the formulation is clear and systematic, the conceptual novelty is limited. The paper extends existing ideas rather than introducing a fundamentally new mechanism.

**4. Theoretical efficiency may not translate to hardware speedup**

Although QAdA reduces algorithmic complexity in theory, its query-dependent masking can create irregular computation patterns that are less compatible with hardware-optimized static sparsity. Modern accelerators often favor predefined or block-sparse layouts. As a result, the practical runtime benefit of QAdA remains uncertain.

**Questions:**

See the weaknesses.

---

> ### Author Response · Authors · 2025-11-19
>
> We thank the reviewer for their helpful and constructive feedback.
>
> >"Comparison limited to oracle and static baselines"
>
> Our paper was not primarily designed as a "method" paper, in which we propose a novel state-of-the-art method for efficient attention computation in long contexts.
> As such, the adaptive algorithm we describe is not intended to be a production-ready solution, but rather a proof of concept that showcases the potential gains achievable through adaptive head selection. As stated in our Limitations section, it is meant to be a foundation for future research rather than a mature solution. This explains the lack of comparison to previous work.
> Instead, our primary goal is scientific discovery and mechanistic understanding: we aim to study the behavior of attention heads in terms of context length and attention distribution. As highlighted in Contribution 1 of the Introduction, our main contribution is a detailed analysis of attention patterns, through a series of experiments on various models and tasks, combined with many ablations. The study of our adaptive method itself contributes to this deeper understanding of attention mechanisms.
> As such an analysis, focused on query-adaptivity, had not been attempted before, we do believe that our contribution is novel.
>
>
> >"Complementarity with query-adaptive sparsity not empirically validated"
> and
> >"Conceptual contribution is modest and primarily integrative"
>
> Our main contribution is not a novel idea, but rather an extensive scientific investigation of attention mechanisms, particularly in the context of query-adaptiveness, and of the possible trade-offs between accuracy and sparsity.
>
> >"Theoretical efficiency may not translate to hardware speedup"
>
> In the Limitations section, we acknowledge the fact that while the studied algorithm is sufficient to showcase the query-adaptive nature of attention and to be used as a support for our experiments, it does not constitute a production-ready method; designing such a method would warrant another article.
> However, while current accelerators do often favor predefined or block-sparse layouts, new accelerators could be designed to support more flexible structures (see, e.g., FlexAttention for an example of this trend https://pytorch.org/blog/flexattention/).

---

### Official Review · Reviewer_MYZu · 2025-10-31

**Soundness:** 2
**Presentation:** 2
**Contribution:** 2
**Rating:** 4
**Confidence:** 4

**Summary:**

This paper presents a Query-Adaptive Attention methodology that dynamically determines whether to locally approximate each attention head for a query when processing long contexts, by predicting the attention score using a Gaussian approximation rather than relying on static labeling. This approach enabled computational cost savings while maintaining performance on downstream tasks.

**Strengths:**

- QAdA can be applied to pretrained language models without modifying their architecture, and it operates directly during inference without any fine-tuning, making it practically efficient for real-world use.
- The paper provides experimental evidence that attention heads switch between local and long-context behavior depending on the query, convincingly demonstrating the necessity of an adaptive approach.
- By using a Gaussian approximation, the method simplifies the computational complexity of the adaptive mechanism, thereby improving its practical applicability.

**Weaknesses:**

- The paper does not provide quantitative evidence of resource savings during the actual inference stage, which weakens the credibility of its efficiency claims. It only presents a theoretical comparison of computational complexity without including experimental results such as GPU memory usage or latency reduction. Although the study emphasizes efficiency improvement, it lacks direct empirical data to support this point.

- The accuracy of head classification based on the Gaussian approximation is insufficiently validated with quantitative results. While Figure 8(a) visualizes FP and FN ratios, concrete metrics such as accuracy or recall are not provided in tabular form. As a result, it is difficult to quantitatively assess how precisely the approximation distinguishes between head types.

- The practical significance of performance improvement relative to the computational overhead introduced by QAdA is not clearly demonstrated. Although the results show that performance remains stable as sparsity is adjusted, there is no experimental analysis of how much additional computational cost QAdA incurs. The balance between efficiency and performance is therefore not adequately evaluated.

- The explanation of sparsity in the Introduction is insufficient, and Figure 1 may misleadingly suggest that sparsity is a directly controllable independent variable. While Figure 1 presents performance variation with sparsity on the x-axis, the paper does not clearly define how sparsity is computed or derived. This could lead readers to misunderstand sparsity as a tunable parameter rather than a derived property.

**Questions:**

- Why does the Abstract not include a summary of the experimental results? The Abstract focuses primarily on conceptual proposals and does not provide a summary of experimental findings or performance metrics. As a result, the practical impact of the study is not immediately clear from the first impression.
- Why are the results presented mainly through sparsity-based performance curves rather than numerical tables?
Most experimental results are shown as performance curves plotted against sparsity. While this visualization effectively illustrates overall trends, it limits direct numerical comparison across models and hinders detailed quantitative analysis.
- According to the time complexity of QAdA shown in Table 1, there is a possibility of computational bottlenecks in models with large dimensionality $d$. Why is there no discussion or analysis regarding this issue?
- QAdA exhibits a conservative classification tendency where the false negative rate is lower than the false positive rate. How do the authors believe this conservative behavior contributes to maintaining model performance?

---

> ### Author Response · Authors · 2025-11-19
>
> We thank the reviewer for their helpful and constructive feedback.
>
> >"The paper does not provide quantitative evidence of resource savings [...] it lacks direct empirical data to support this point."
>
> and
>
> >"The practical significance of performance improvement relative [...]The balance between efficiency and performance is therefore not adequately evaluated."
>
> Our paper was not primarily designed as a "method" paper, in which we propose a novel state-of-the-art method for efficient attention computation in long contexts.
> As such, the adaptive algorithm we describe is not intended to be a production-ready solution, but rather a proof of concept that showcases the potential gains achievable through adaptive head selection. As stated in our Limitations section, it is meant to be a foundation for future research rather than a mature solution.
> This explains, in particular, the lack of a compute-efficient implementation. Designing one would warrant a separate article.
> Instead, our primary goal is scientific discovery: we aim to study the behavior of attention heads in terms of context length and attention distribution. As highlighted in Contribution 1 of the Introduction, our main contribution is a detailed analysis of attention patterns, through a series of experiments on various models and tasks, combined with many ablations. The study of our adaptive method itself contributes to this deeper understanding of attention mechanisms.
>
> >"The explanation of sparsity in the Introduction is insufficient, and Figure 1 may misleadingly suggest that sparsity is a directly controllable independent variable. While Figure 1 presents performance variation with sparsity on the x-axis, the paper does not clearly define how sparsity is computed or derived. This could lead readers to misunderstand sparsity as a tunable parameter rather than a derived property."
>
> Our intention in this specific Figure was to visually link the two variables with a direct practical impact, namely sparsity and accuracy, and to abstract away the threshold, which is of no intrinsic interest.
> Nonetheless, we agree with you that this presents a risk of confusion; we will add additional explanations. Thank you for this advice.
>
> >"Why does the Abstract not include a summary of the experimental results? The Abstract focuses primarily on conceptual proposals and does not provide a summary of experimental findings or performance metrics. As a result, the practical impact of the study is not immediately clear from the first impression."
>
> Thank you! We will make this clearer in the revised version.
>
> >"Why are the results presented mainly through sparsity-based performance curves rather than numerical tables? Most experimental results are shown as performance curves plotted against sparsity. While this visualization effectively illustrates overall trends, it limits direct numerical comparison across models and hinders detailed quantitative analysis."
>
> We will include the data corresponding to some of the main curves as tables in the Appendix for increased clarity.
>
> >"According to the time complexity of QAdA shown in Table 1, there is a possibility of computational bottlenecks in models with large dimensionality $d$. Why is there no discussion or analysis regarding this issue?"
>
> The internal dimension $d$ of current state-of-the-art models is on the order of a few thousand, whereas long contexts (those for which sparsity methods become enticing) tend to be at least several times larger. As such, this is not a concern with current architectures and use cases.
> We will add a footnote regarding this point.
>
>
> >"QAdA exhibits a conservative classification tendency where the false negative rate is lower than the false positive rate. How do the authors believe this conservative behavior contributes to maintaining model performance?"
>
> Indeed, for the same given value of the threshold, QAdA classifies more heads as long context than the oracle. However, the variable of interest is the degree of sparsity, rather than the threshold parameter itself; as such, for a fair comparaison, one should compare the oracle and QAdA at equal sparsity levels (rather than at equal threshold levels), which is what we always show.
>
> Note also that the ratio of false positives to false negatives is not always greater than 1 (as seen in Figure 8). Overall, we interpret this ratio as a consequence of the difference in shape between our chosen approximation and the true underlying distribution; a different choice of approximation would result in a different error profile.

---

### Official Review · Reviewer_XCVq · 2025-11-01

**Soundness:** 3
**Presentation:** 3
**Contribution:** 3
**Rating:** 8
**Confidence:** 3

**Summary:**

This paper studies which attention heads actually need global context on a per-query basis and proposes QAdA, a lightweight criterion that decides, at each token, whether a head can be approximated with a local window (attention sinks + local tokens) or must attend to the full sequence. The method estimates the bulk attention mass without computing bulk scores by modeling the distribution of bulk keys as Gaussian, which makes the bulk softmax term log-normal in closed form.

**Strengths:**

- The paper is very well written and addresses an important problem in attention head pruning. I genuinely think that this is a strong contribution.
- The method proposed is novel and interesting, and is also theoretically grounded. Rather than assigning a fixed “local vs global” label to each head, it decides this adaptively at the query level using a simple analytical model, which I found very interesting.
- The empirical results favor the method. In particular, the paper evaluates QAdA across multiple long-context benchmarks (RULER, LongBench) and two challenging long-context variants. Performance–sparsity curves indicate that while static criteria remain safe up to roughly ~60% local heads, QAdA maintains accuracy at ~80–90% sparsity and, in some cases even exceeds dense-attention baselines, suggesting that adaptivity can reduce harmful attention to irrelevant context.

**Weaknesses:**

- Although this is somewhat of a shallow comment, I think that the fact that this is tested on 7B models could be somewhat of a limitation. Recent work (Barbero et al., 2025) shows that model size heavily affects the formation of attention sinks. As such, it would be nice to see how these results generalize to smaller and larger models. I understand if the authors are unable to test very big models due to computational requirements.

- The authors take the first 16 tokens as sink tokens, this behavior typically appears only in the first one. Is there any reason why the authors made this choice?

Barbero, Federico, et al. "Why do LLMs attend to the first token?." arXiv preprint arXiv:2504.02732 (2025).

**Questions:**

- Could the authors elaborate on the rationale for modeling bulk keys as Gaussian? In practice, how well does this approximation fit the observed key/score distributions across layers/heads, and what diagnostics did you use to decide it (e.g., moment fit, error of the log-mass estimate, QQ plots)? Please also comment on when this approximation breaks down and how sensitive QAdA is to such deviations.
- Do the authors have some intuition of which layers are pruned the most? Does pruning happen uniformly across layers?

---

> ### Author Response · Authors · 2025-11-19
>
> We are very thankful to the reviewer for their positive and encouraging feedback.
>
> >"Although this is somewhat of a shallow comment, I think that the fact that this is tested on 7B models could be somewhat of a limitation. Recent work (Barbero et al., 2025) shows that model size heavily affects the formation of attention sinks. As such, it would be nice to see how these results generalize to smaller and larger models. I understand if the authors are unable to test very big models due to computational requirements."
>
> This is a reasonable concern. Given our compute budget, we chose to run more experiments on a variety of mid-sized models and tasks, rather than to run fewer experiments on larger models. Smaller models are less interesting, as they tend to have poor performance on long context tasks, to the point where the experiments become almost meaningless.
>
> >"The authors take the first 16 tokens as sink tokens, this behavior typically appears only in the first one. Is there any reason why the authors made this choice?"
>
> Our very early preliminary experiments suggested that it could lead to slightly increased stability; it also meant an almost negligible increase in computational cost compared to using a single sink token. We did not systematically optimize the number of sink tokens, which we leave to future works.
>
> >"Could the authors elaborate on the rationale for modeling bulk keys as Gaussian? In practice, how well does this approximation fit the observed key/score distributions across layers/heads, and what diagnostics did you use to decide it (e.g., moment fit, error of the log-mass estimate, QQ plots)? Please also comment on when this approximation breaks down and how sensitive QAdA is to such deviations"
>
> Our main objective was to prove and study the query-adaptive nature of attention. As such, we did not need the best possible statistical test (and associated approximation): it only needed to be good enough for the studied phenomena to be clearly visible, which the Gaussian-based test was.
> We picked it among other candidates because approximating distributions with a Gaussian distribution with the same first two moments is a common trick (e.g. PCA), but as emphasized in the last paragraph of Section 3.1, we expect that further gains could be made by optimizing the choice of approximating distribution.
>
> Regarding the robustness, we have observed that the Gaussian approximation is in general NOT a very good fit of the key distribution (see e.g. Appendix E), but that the resulting statistical test itself nonetheless performs well, as showcased throughout the paper. Hence the query-adaptive algorithm seems fairly robust to the quality of the approximation.
>
> >"Do the authors have some intuition of which layers are pruned the most? Does pruning happen uniformly across layers?"
>
> This is an interesting question. We investigate exactly this question in Figure 6, where we show that all layers contribute to the long-context understanding of the models, and all layers have heads that can be pruned. Interestingly, we find that for tasks where the context is redundant, both the oracle and QAdA prune all heads after the first head. This is an intersting and important observation, as it shows that we can correctly identify cases where there is no need to attend to the long context.

---

### Official Review · Reviewer_6y2m · 2025-11-01

**Soundness:** 3
**Presentation:** 3
**Contribution:** 3
**Rating:** 4
**Confidence:** 4

**Summary:**

This paper studies how attention heads in large language models (LLMs) process long-context information and proposes a method to identify which heads need access to the full context versus only local information. The authors observe that while some attention heads consistently attend to local tokens, others adaptively switch between local and long-context attention depending on the query. They introduce QAdA (Query-Adaptive Attention), a computationally efficient criterion that predicts which heads require long-context information using only local key statistics and a Gaussian approximation of bulk attention scores based on second-order moments. Through experiments on Llama, Qwen, and Mistral models across benchmarks like RULER and LongBench, they demonstrate that their query-adaptive approach can label up to 90% of heads as local (compared to 60% for static methods) while maintaining performance, and in some cases even improving upon standard full attention. The method achieves near-oracle performance with significantly lower computational cost, advancing both the efficiency and mechanistic understanding of attention mechanisms in long-context settings.

**Strengths:**

1. The paper provides an extensive analysis showing that attention heads dynamically switch between local tokens and long-context ones. It demonstrates that a query-adaptive oracle criterion can achieve a substantially higher percentage of sparsity (up to 90%) by labeling heads as local, compared to static criteria (up to 60%), all while maintaining downstream task performance.

2. The paper introduces a novel and efficient query-adaptive attention criterion, QAdA, based on second-order statistics (mean and covariance) of the attention scores. This method can predict which attention heads need long-context information without computing the full attention scores. This design gives QAdA a constant run-time complexity with respect to the total sequence length for the head selection decision.

3. The QAdA criterion outperforms static criteria and achieve near-oracle performance on a variety of long-context benchmarks (RULER, LongBench) across three popular families of Large Language Models (LLMs): Llama, Qwen, and Mistral. Furthermore, in certain cases, the query-adaptive approach was observed to surpass the baseline performance of standard full attention, suggesting benefits beyond just efficiency by effectively pruning unnecessary or irrelevant context.

**Weaknesses:**

1. QAdA assumes that the key embeddings in the non-local position follow a Gaussian distribution, and the paper lacks theoretical proof or comprehensive empirical evidence to validate this assumption. Furthermore, Figure 2 illustrates the actual bulk distribution (gray), but it seems that their distributions are somewhat not close to the Gaussian distribution.

2. While QAdA is designed to be efficient during inference, it requires an initial calibration step. This step involves calculating the mean and covariance of the attention scores for a specific, representative subset of the training data. This introduces an additional, potentially non-trivial, computational overhead during the pre-processing phase before the model can be effectively used for long-context inference.

3. QAdA hinges on the choice of the threshold parameter $\tau$. This threshold determines how strictly a head is classified as needing long-context access based on its predicted long-context probability. Since the optimal $\tau$ must be empirically determined, likely via a complex search over a small validation set, the model's robust performance is highly sensitive to this manual hyperparameter tuning.

**Questions:**

1. What is the theoretical justification of Equation 4/5? or is there any empirical evidence of this? And how robust is the method if the true distribution is non-Gaussian?

2. What is the precise computational overhead of this calibration step? (e.g., relative latency to the full training or fine-tuning process), and how does this overhead impact the overall practical efficiency gains, especially for models deployed in dynamic environments?

3. The performance of QAdA can be sensitive to the threshold parameter ($\tau$). How the threshold parameter can be chosen? Is there any process that is more robust, adaptive, or automatic mechanism for setting this threshold to ensure performance stability across different models, tasks, and data distributions without relying on a complex, manual search process?


Minor issues:
- Paragraph title format is not consistent – some contains a comma, others are not.
- Equation (3) would be clearer to write either: E_{k^T ~ ν_bulk} [exp(qk^T / √d)] or E_{k_i ~ ν_bulk} [exp(qk_i^T / √d)] (if using subscript i)
- In equation (4), what does (N(\mu_K, \Sigma_K))^T mean? A random variable k^T from \nu^{bulk} seems a vector, and N(\mu_K, \Sigma_K) is a distribution on vectors. But, what is the support of distribution (N(\mu_K, \Sigma_K))^T?

---

> ### Author Response · Authors · 2025-11-19
>
> We thank the reviewer for their careful reading and constructive feedback.
>
> > "What is the theoretical justification of Equation 4/5? or is there any empirical evidence of this? "
>
> As detailed in the last paragraph of Subsection 3.1, we do not expect the Gaussian approximation used to be the best approximation one could find, or even to be particularly good (as this is not a method paper, we do not aim for optimality).
> Rather, our main purpose is to show the query-adaptive nature of attention. As such, it suffices to prove attention is query-adaptive for this (possibly suboptimal) approximation method: a better approximation would only further reinforce the conclusion.
>
> >"And how robust is the method if the true distribution is non-Gaussian?"
> In fact, the true distribution appears to be quite far from being Gaussian, as discussed in Appendix E, and the method is nonetheless robust.
>
>
> >"What is the precise computational overhead of this calibration step? (e.g., relative latency to the full training or fine-tuning process), and how does this overhead impact the overall practical efficiency gains, especially for models deployed in dynamic environments?"
>
> The exact cost would depend on implementation details, but as a rule of thumb, a single forward pass on a long sequence should be enough for calibration. In a typical deployment setting (e.g. an LLM serving as a chatbot or an assistant), the model, once calibrated, would be queried thousands to millions of times, making the cost of the initial calibration (which can be framed as a very lightweight post-training technique) negligible.
>
>
> >"The performance of QAdA can be sensitive to the threshold parameter. How the threshold parameter can be chosen? Is there any process that is more robust, adaptive, or automatic mechanism for setting this threshold to ensure performance stability across different models, tasks, and data distributions without relying on a complex, manual search process?"
>
> This is a limitation of all such methods (including pre-existing static methods), for which one must necessarily decide on an acceptable trade-off between accuracy and sparsity. A reasonable solution would be to calibrate the threshold on an array of typical tasks (and for a tolerated level of accuracy loss); as argued above, the cost of such a procedure would be fairly negligible compared to the cost of large scale deployment.
>
> >"In equation (4), what does (N(\mu_K, \Sigma_K))^T mean? [...]"
>
> $\mathcal N(\mu_K, \Sigma_K)$ is a Gaussian distribution in $\mathbb{R}^d$, where $d$ is the dimension of the key vectors. $\left(\mathcal  N(\mu_K, \Sigma_K)  \right)^{T_\text{bulk}}$ is the independent product of $T_\text{bulk}$ such distributions; hence $\nu^{\text{bulk}} \approx \left(\mathcal  N(\mu_K, \Sigma_K)  \right)^{T_\text{bulk}}$ is a random variable in $(\mathbb{R}^d)^{T_\text{bulk}} = \mathbb{R}^{d\cdot T_\text{bulk}}$. Do you think that replacing the current notation by $\left(\mathcal  N(\mu_K, \Sigma_K)  \right)^{\otimes T_\text{bulk}}$ would make it clearer?

---

### Meta-Review · Area_Chair_gr2m · 2025-12-29

**Summary:**

This paper studies how attention heads in large language models (LLMs) process long-context information and proposes a method to identify which heads need access to the full context versus only local information. This submission receives 4 reviews, and 3 of them are negative. The main concern is that the paper does not provide quantitative evidence of resource savings during the actual inference stage, which weakens the credibility of its efficiency claims. It only presents a theoretical comparison of computational complexity without including experimental results such as GPU memory usage or latency reduction. Although the study emphasizes efficiency improvement, it lacks direct empirical data to support this point. The authors, however, fail to address this concern by stating that "Our paper was not primarily designed as a 'method' paper, in which we propose a novel state-of-the-art method for efficient attention computation in long contexts. As such, the adaptive algorithm we describe is not intended to be a production-ready solution, but rather a proof of concept that showcases the potential gains achievable through adaptive head selection."

**Reviewer Scores:**

NA

---

### Decision · Program_Chairs · 2026-01-26

Reject